# Voting-based Approaches for Differentially Private Federated Learning

## Abstract

Differentially Private Federated Learning (DPFL) is an emerging field with many
applications. Gradient averaging based DPFL methods require costly commu-
nication rounds and hardly work with large-capacity models, due to the explicit
dimension dependence in its added noise. In this paper, inspired by the non-
federated knowledge transfer privacy learning methods, we design two DPFL
algorithms (*AE-DPFL* and *kNN-DPFL*) that provide provable DP guarantees for
both instance-level and agent-level privacy regimes. By voting among the data
*labels* returned from each local model, instead of averaging the gradients, our
algorithms avoid the dimension dependence and significantly reduces the commu-
nication cost. Theoretically, by applying secure multi-party computation, we could
exponentially amplify the (data-dependent) privacy guarantees when the margin
of the voting scores are distinctive. Empirical evaluation on both instance and
agent level DP is conducted across five datasets. When aligning privacy cost the
same, we show $2\%$ to $12\%$ higher accuracy compared to DP-FedAvg, or aligning
accuracy the same, we show that less than $65\%$ privacy cost is achieved.

## 1 Introduction

Federated learning (FL) [McMahan et al., 2017, Bonawitz et al., 2017b, Mohassel and Zhang, 2017,
Smith et al., 2017] is an emerging paradigm of distributed machine learning with a wide range of
applications [Kairouz et al., 2019]. FL allows distributed agents to collaboratively train a centralized
machine learning model without sharing each of their local data, thereby sidestepping the ethical and
legal concerns that arise in collecting private user data for the purpose of building machine-learning
based products and services.

The workflow of FL is often enhanced by secure multi-party computation [Bonawitz et al., 2017b]
(MPC) so as to handle various threat models in the communication protocols, which provably ensures
that agents can receive the output of the computation (e.g., the sum of the gradients) but nothing in
between (e.g., other agents' gradients).

However, MPC alone does not protect the agents or their users from inference attacks that use only
the output, or combine the output with auxiliary information. Extensive studies demonstrate that
these attacks may lead to a blatant reconstruction of proprietary datasets [Dinur and Nissim, 2003],
high-confidence identification of individuals (a legal liability for the participating agents) [Shokri
et al., 2017], or even completion of social security numbers [Carlini et al., 2019]. Motivated by
these challenges, there have been a number of recent efforts [Truex et al., 2019, Geyer et al., 2017,
McMahan et al., 2018] in developing federated learning methods with differential privacy (DP)
[Dwork et al., 2006], which is a well-established definition of privacy that provably prevents such
attacks.

Existing methods in differentially private federated learning (DPFL), e.g., DP-FedAvg [Geyer et al.,
2017, McMahan et al., 2018] and the recent state-of-the-art DP-FedSGD [Truex et al., 2019], are

predominantly noisy gradient based methods, which build upon the NoisySGD method, a classical algorithm in (non-federated) DP learning [Song et al., 2013, Bassily et al., 2014, Abadi et al., 2016]. They work by iteratively aggregating (multi-)gradient updates from individual agents using a differentially private mechanism. A notable limitation for this approach is that they require clipping the $\ell_2$ magnitude of gradients to a threshold $S$ and adding noise proportional to $S$ to *every coordinate* of the high dimensional parameters from the shared global model. The clipping and perturbation steps introduce either large bias (when $S$ is small) or large variance (when $S$ is large), which interferes with convergence of SGD, which makes scaling to large-capacity models difficult. In Sec. A, we concretely demonstrate these limitations with examples and theory. Particularly, we show that FedAvg may fail to decrease the loss function using gradient clipping, and DP-FedAvg requires many outer-loop iterations (i.e., many rounds of communication to synchronize model parameters) to converge under differential privacy.

In this paper, we consider a fundamentally different DP learning setting known as the *Knowledge Transfer* model [Papernot et al., 2017] (a.k.a. the *Model-Agnostic Private learning* model [Bassily et al., 2018]). This model requires an *unlabeled* dataset to be available *in the clear*, which makes this setting slightly more restrictive. However, when such a public dataset is indeed available (it often is in federated learning with domain adaptation, see, e.g., Peterson et al. [2019], Mohri et al. [2019], Peng et al. [2019b]), it could substantially improve the privacy-utility tradeoff in DP learning [Papernot et al., 2017, 2018, Zhu et al., 2020].

The goal of this paper is to develop DPFL algorithms under the *knowledge transfer* model, for which we propose two algorithms (*AE-DPFL* and *kNN-DPFL* ), that further develop from the *non-distributed* Private-Aggregation-of-Teacher-Ensembles (PATE) [Papernot et al., 2018] and Private-kNN [Zhu et al., 2020] to the FL setting. We discover that the distinctive characteristics of these algorithms make them *natural* and *highly desirable* for DPFL tasks. Specifically, the private aggregation is now essentially privately releasing "ballot counts" in the (one-hot) label space, instead of the parameter (gradient) space. This naturally avoids the aforementioned issues associated with high dimensionality and gradient clipping. Instead of transmitting the gradient update, transmitting the vote of the "ballot counts" tremendously reduce the communication cost. Moreover, many iterations of the model update using noise addition with SGD, leads to poor privacy guarantee, where our methods exactly avoid this and use voting on labels, thus significantly outperform the state-of-the-art DPFL methods.

Our contributions are summarized in four folds.

1. We construct examples to demonstrate that DP-FedAvg (a) may fail due to gradient clipping and (b) requires many rounds of communications (see Section Challenge in the appendix); while our approach naturally avoids both limitations.
2. We design two voting-based distributed algorithms that provide provable DP guarantees on both *agent-level* and *instance (of-each-agent)-level* granularity, which makes them suitable for both well-studied regimes of FL: (a) distributed learning from on-device data; (b) collaboration of a few large organizations.
3. We demonstrate "privacy-amplification by ArgMax" by a new MPC technique [Dery et al., 2019] — our proposed private voting mechanism enjoys an *exponentially stronger* (data-dependent) privacy guarantee when the "winner" wins by a large margin.
4. Extensive evaluation demonstrates that our method systematically improves the privacy-utility trade-off over DP-FedAvg and DP-FedSGD, and that our methods are more robust towards distribution-shifts across agents.

**A remark of our novelty.** Though *AE-DPFL* and *kNN-DPFL* are algorithmically similar to the original *PATE* [Papernot et al., 2018] and *Private-KNN* [Zhu et al., 2020], they are not the same and we facilitate them to a new problem — *federated learning*. The facilitation itself is nontrivial and requires substantial technical innovations. We highlight three challenges below.

To begin with, several key DP techniques that contribute to the success of PATE and Private-kNN in the standard settings are no longer applicable (e.g., privacy amplification by sampling and noisy screening). This is partially because in standard private learning, the attacker only sees the final models; but in FL, the attacker can eavesdrop in all network traffic and could be a subset of the agents themselves.

Moreover, PATE and Private-kNN only provide instance-level DP. We show *AE-DPFL* and *kNN-DPFL* also satisfy the stronger agent-level DP. *AE-DPFL*'s agent-level DP parameter is, interestingly,

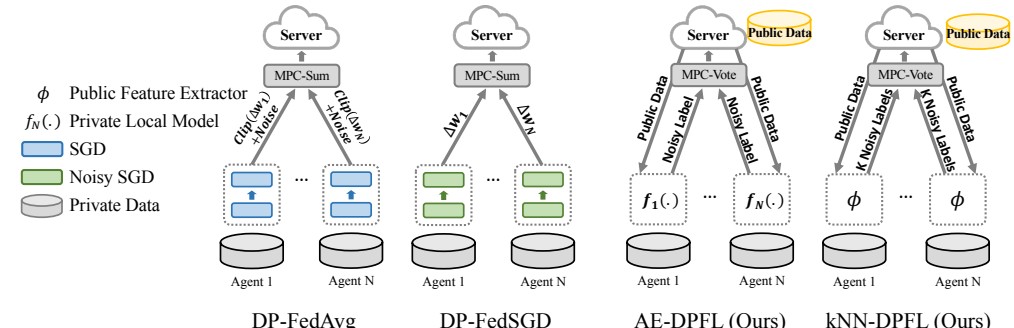

Figure 1: The structural difference of our methods to DP-FedAvg and DP-FedSGD. DP-FedAvg and *AE-DPFL* are for agent-level DP. DP-FedSGD and *kNN-DPFL* are for instance-level DP.

a factor of 2 better than its instance-level DP parameter. *kNN-DPFL* in addition enjoys a factor of $k$ amplification for the instance-level DP.

Thirdly, a key challenge of FL is data heterogeneity of individual agents. Methods like PATE randomly split the dataset so each teacher is identically distributed, but this assumption is violated with heterogeneous agents. Similarly, methods like Private-kNN have also been demonstrated only under homogeneous settings. In contrast, our proposed methods – AE-DPFL and kNN-DPFL – exhibit robustness to data heterogeneity and domain shifts, as demonstrated in our experiments. Note that techniques like domain adaptation may lead to further complementary benefits, but we defer its exploration to future work, while focusing our scope here on novel techniques for DPFL.

## 2 Preliminary

Differential privacy [Dwork et al., 2006] is a quantifiable definition of privacy that provides provable guarantees against identification of individuals in a private dataset.

**Definition 1.** *Differential Privacy: A randomized mechanism $\mathcal{M} : \mathcal{D} \rightarrow \mathcal{R}$ with a domain $\mathcal{D}$ and range $\mathcal{R}$ satisfies $(\epsilon, \delta)$-differential privacy, if for any two* adjacent *datasets $D, D' \in \mathcal{D}$ and for any subset of outputs $O \subseteq \mathcal{R}$, it holds that $\Pr[\mathcal{M}(D) \in O] \leq e^\epsilon \Pr[\mathcal{M}(D') \in O] + \delta$.*

The definition indicates that one could not distinguish between $D$ and $D'$ therefore protecting the "delta" between $D, D'$. Depending on how *adjacency* is defined, this "delta" comes with different semantic meaning. We consider two levels of granularity:

**Definition 2.** *Agent-level DP: When $D'$ is constructed by adding or removing an agent from $D$ (with all data points from that agent).*

**Definition 3.** *Instance-level DP: When $D'$ is constructed by adding or removing one data point from any of the agents.*

The above two definitions are each important in particular situations. For example, when a smart phone app jointly learns from its users' text messages, it is more appropriate to protect each user as a unit, which is agent-level DP. In another situation, when a few hospitals would like to collaborate on a patient study through federated learning, obfuscating the entire dataset from one hospital is meaningless, which makes instance-level DP better-suited to protect an individual patient from being identified.

**DPFL Baselines:** DP-FedAvg [Geyer et al., 2017, McMahan et al., 2018] (Figure 1), a representative DPFL algorithm, when compared to FedAvg, it enforces clipping of per-agent model gradient to a threshold $S$ and adds noise to the scaled gradient before it is averaged at the server, which ensures agent-level DP. DP-FedSGD [Truex et al., 2019, Peterson et al., 2019], is one of the state-of-the-arts that focus on instance-level DP. It performs NoisySGD [Abadi et al., 2016] for a fixed number of iterations at each agent. The gradient updates are averaged on each communication round at the server, as shown in Figure 1.

## 3 Our Approach

We propose two voting-base algorithms, termed aggregation ensemble DPFL "*AE-DPFL*" and k Nearest Neighbor DPFL "*kNN-DPFL*". Each algorithm first privately labels a subset of data from the server and then trains a global model using pseudo-labeled data.

### 3.1 Aggregation Ensemble - DPFL

In *AE-DPFL* (Algorithm 1), each agent $i$ trains a local agent model $f_i$ using its own private local data. The local model is never revealed to the server but only used to make predictions for unlabeled data (queries). For each query $x_t$, every agent $i$ adds Gaussian Noise to the prediction (i.e., $C$-dimensional histogram where each bin is zero except the $f_i(x_t)$-th bin is 1). The "pseudo label" is achieved with the majority vote returned by aggregating the noisy predictions from the local agents.

For instance-level DP, the spirit of our method shares with PATE, in the aspect of by adding or removing one instance, it can *change* at most one agent's prediction. The same argument also naturally applies to *adding or removing one agent*. In fact we gain a factor of 2 in the stronger agent-level DP due to a smaller sensitivity in our approach (see Theorem 4).

Another important difference is that in the original PATE, the teacher models are trained on I.I.D data (random splits of the whole private data), while in our case, the agents are naturally present with different distributions. We propose to optionally use domain adaptation techniques to mitigate these differences when training the agents.

### 3.2 kNN - DPFL

From Definition 2 and 3, preserving agent-level DP is generally more difficult than the instance-level DP. We find that for *AE-DPFL*, the privacy guarantee for instance-level DP is weaker than its agent-level DP guarantee (see Theorem 4). To amplify the instance-level DP, we now introduce our *kNN-DPFL*.

As in Algorithm 2, each agent maintains a data-independent feature extractor $\phi$, i.e., an ImageNet [Deng et al., 2009] pre-trained network without the classifier layer. For each unlabeled query $x_t$, agent $i$ first finds the $k_i$ nearest neighbors to $x_t$ from its local data by measuring the Euclidean distance in the feature space $\mathcal{R}^{d_\phi}$. Then, $f_i(x_t)$ outputs the frequency vector of the votes from the nearest neighbors, which equals to $\frac{1}{k}(\sum_{j=1}^{k} y_j)$, where $y_j \in \mathcal{R}^C$ indicates the one-hot vector of the ground-truth label. Subsequently, $\tilde{f}_i(x_t)$ from all agents are privately aggregated with the argmax of the noisy voting scores returned to the server.

Our kNN-DPFL differs from Private-kNN in that we apply kNN on each agent's local data instead of the entire private dataset. This distinction together with MPC allows us to receive up to $kN$ neighbors while bounding the contribution of individual agents by $k$. Comparing to *AE-DPFL*, this approach enjoys a stronger instance-level DP guarantee since the sensitivity from adding or removing one instance is a factor of $k/2$ times smaller than that of the agent-level (see the proof in Theorem 4).

**How to implement MPC-vote?** Dery et al. [2019] assumes a set of (honest and non-colluding) external entities, named talliers, $\mathbb{T} = \{T_1, ..., T_J\}$. Then, each agent applies secret sharing for creating $J$ shares of the private ballots ($\tilde{f}_i(x_t)$ in our case), and distributing them among the $J$ talliers. After receiving the ballot shares from all agents, the tallier will compute the sum of share vectors and find the index $y \in \{1, ..., C\}$ with the highest scores and send that to the server. We refer the reader to Protocol 1 in Dery et al. [2019] for a detailed procedure. We highlight that using MPC-vote (only the top-one index is revealed to the server) instead of MPC-sum results in a stronger differential privacy guarantee, as discussed in the next section.

### 3.3 Privacy Analysis

Our privacy analysis is based on Renyi differential privacy (RDP) [Mironov, 2017]. We defer the background about RDP, its connection to DP and all proofs of our technical results to the appendix RDP section.

**Theorem 4** (Privacy guarantee). *Let AE-DPFL and kNN-DPFL answer $Q$ queries with noise scale $\sigma$. For agent-level protection, both algorithms guarantee $(\alpha, \frac{Q\alpha}{2\sigma^2})$-RDP for all $\alpha \geq 1$. For instance-level protection, AE-DPFL and kNN-DPFL obey $(\alpha, \frac{Q\alpha}{\sigma^2})$ and $(\alpha, \frac{Q\alpha}{k\sigma^2})$-RDP respectively.*

**Remark 1.** *Theorem 4 suggests that both algorithms achieve agent-level and instance-level differential privacy. With the same noise injection to the agent's output, kNN-DPFL enjoys a* stronger *instance-level DP (by a factor of $k/2$) compared to its agent-level guarantee, while AE-DPFL's instance-level DP is* weaker *by a factor of 2. Since AE-DPFL allows an easy-extension with the domain adaptation technique, we choose to use AE-DPFL for the agent-level DP and apply kNN-DPFL for the instance-level DP in the experiments.*

| **Algorithm 1** *AE-DPFL* with MPC-Vote | **Algorithm 2** *kNN-DPFL* with MPC-Vote |
|---|---|
| **input** Noise level $\sigma$, unlabeled public data $\mathcal{D}_G$, integer $Q$. | **input** Noise level $\sigma$, unlabeled public data $\mathcal{D}_G$, integer $Q$, feature map $\phi$. |
| 1: Train local model $f_i$ using $\mathcal{D}_i$ or using $(\mathcal{D}_i, \mathcal{D}_G)$ with any domain adaptation techniques. | 1: **for** $t = 0, 1, ..., Q$, pick $x_t \in \mathcal{D}_G$ **do** |
| 2: **for** $t = 0, 1, ..., Q$, pick $x_t \in \mathcal{D}_G$ **do** | 2:   **for** each agent $i$ in $1, ..., N$ (in parallel) **do** |
| 3:   **for** each agent $i$ in $1, ..., N$ (in parallel) **do** | 3:     Apply $\phi$ on $\mathcal{D}_i$ and $x_t$ |
| 4:     $\tilde{f}_i(x_t) = f_i(x_t) + \mathcal{N}(0, \frac{\sigma^2}{N}I_C)$. | 4:     $y_1, ..., y_k \leftarrow$ labels of the k nearest neighbor. |
| 5:   **end for** | 5:     $\tilde{f}_i(x_t) = \frac{1}{k}(\sum_{j=1}^{k} y_j) + \mathcal{N}(0, \frac{\sigma^2}{N}I_C)$ |
| 6:   $\tilde{y}_t = \text{argmax}_{y \in \{1,...,C\}}[\sum_{i=1}^{N} \tilde{f}_i(x_t)]_y$ via MPC. | 6:   **end for** |
| 7: **end for** | 7:   $\tilde{y}_t = \text{argmax}_{y \in \{1,...,C\}}[\sum_{i=1}^{N} \tilde{f}_i(x_t)]_y$ via MPC. |
| | 8: **end for** |
| **output** A global model $\theta$ trained using $(x_t, \tilde{y}_t)_{t=1}^{Q}$ | **output** A global model $\theta$ trained using $(x_t, \tilde{y}_t)_{t=1}^{Q}$ |

**Communication Cost:** Finally, we find that our methods are *embarrassingly parallel* as each agent work independently without any synchronization. Overall, we reduce the (per-agent) up-stream communication cost from $d \cdot T$ floats (model size times $T$ rounds) to $C \cdot Q$, where $C$ is number of classes and $Q$ is the number of data points. Moreover, the communication overheads due to MPC protocols approach a multiplicative constant over the transmitted data for both MPC-sum and MPC-vote ([Bonawitz et al., 2017a, Dery et al., 2019]).

## 4 Experimental Results

In this section, we apply our *AE-DPFL* for agent-level DP and *kNN-DPFL* for instance-level DP based on their distinctive characteristics in privacy guarantee.

### 4.1 Agent-level DP Evaluation

To investigate various heterogeneous scenarios, we consider: (1) non-I.I.D partition of local data (MNIST); (2) data across agents and the server are drawn from different domains (Digit Datasets).

| Datasets | # Agents | Methods | Accuracy (%) | $\epsilon$ |
|---|---|---|---|---|
| | | FedAvg | $97.8 \pm 0.1$ | - |
| MNIST (non-I.I.D) | 100 | DP-FedAvg | $84.2 \pm 0.2$ | 4.3 |
| | | *AE-DPFL* (Ours) | $\mathbf{86.1 \pm 0.2}$ | 4.3 |
| | | FedAvg | $87.6 \pm 0.1$ | - |
| SVHN, MNIST | 200 | FedAvg+DA | $86.9 \pm 0.1$ | - |
| | | DP-FedAvg | $76.3 \pm 0.3$ | 3.7 |
| | | DP-FedAvg+DA | $71.2 \pm 0.4$ | 3.6 |
| $\rightarrow$ USPS (non-I.I.D) | | *AE-DPFL* (Ours) | $83.8 \pm 0.2$ | 3.6 |
| | | *AE-DPFL+DA* (Ours) | $\mathbf{92.5 \pm 0.2}$ | $\mathbf{2.8}$ |

Table 1: **Agent-level DP Evaluation.** We set $\delta = 10^{-3}$ for all datasets. For MNIST, each local agent is with 6 digits. Different local agents do not share exactly the same 6 digits, which is a non-I.I.D setting. Further, we assign SVHN and MIST for local agents and USPS for the server, which is a typical non-I.I.D with domain shift setting.

**MNIST Dataset with Non-I.I.D Partition:** We choose a similar experimental setup as the original FedAvg [McMahan et al., 2017] and DP-FedAvg [Geyer et al., 2017] did. We divide the training set of the sorted MNIST into 100 agents, such that each agent will have samples from 6 digits only. This way, each agent gets 600 data points from 6 classes. We split 30% of the testing set in MNIST as the available unlabeled public data and the remaining testing set used for testing.

**Digit Datasets Evaluation**: MNIST, SVHN and USPS are put together termed as Digit datasets [LeCun et al., 1998, Netzer et al., 2011]. It is a controlled setting to mimic the real situations, where distribution of agent-to-server or agent-to-agent can be different. Based on the size of each dataset,

| Network | Methods | $A, C, D \rightarrow W$ (Acc. %) | $\epsilon$ | $A, C, W \rightarrow D$(Acc.) | $\epsilon$ |
|---|---|---|---|---|---|
| | FedAvg | $90.5 \pm 0.1$ | - | $96.8 \pm 0.1$ | - |
| | DP-FedAvg | $28.1 \pm 0.7$ | 46.6 | $48.2 \pm 0.8$ | 47.1 |
| AlexNet | DP-FedSGD | $32.6 \pm 0.9$ | 4.1 | $48.3 \pm 0.9$ | 4.0 |
| | DP-FedSGD | $75.2 \pm 0.5$ | 12.4 | $83.7 \pm 0.6$ | 7.9 |
| | *kNN-DPFL* ($\sigma = 15$, Ours) | $\mathbf{75.4 \pm 0.3}$ | **3.9** | $\mathbf{84.3 \pm 0.3}$ | **3.7** |
| | FedAvg | $96.5 \pm 0.1$ | - | $97.8 \pm 0.1$ | - |
| ResNet50 | DP-FedSGD | $25.8 \pm 0.6$ | 4.0 | $42.7 \pm 0.5$ | 3.9 |
| | *kNN-DPFL* ($\sigma = 25$, Ours) | $\mathbf{86.3 \pm 0.4}$ | **2.8** | $\mathbf{91.9 \pm 0.2}$ | **2.0** |

Table 2: **Instance-level DP on Office-Caltech dataset for non-I.I.D setting. Total number of local agents is 3. We set $\delta = 10^{-4}$.**

we simulate 140 agents using SVHN with 3000 records each and 60 agents using MNIST with 1000 records each. We split 3000 unlabeled records from USPS at server and the rest data is used for testing.

We notice that DP-FedAvg and FedAvg never see the server distribution. To boost those two algorithms, we further apply a standard domain adaptation (DA) technique — adversarial training [Ganin et al., 2016] on top, denoted as DP-FedAvg+DA and FedAvg+DA, respectively. As a consequence, their local training involves both local data and unlabeled data from the server. Similarly, we define *AE-DPFL+DA* as the DA extension of *AE-DPFL*, where each teacher (agent) model is trained with the same DA technique as that in DP-FedAvg+DA.

In Table 1, we observe that when the privacy cost $\epsilon$ of DP-FedAvg and *AE-DPFL* is close, our method significantly improves the accuracy from 76.3% to 83.8%. (2) The further improved accuracy 92.5% of *AE-DPFL+DA* demonstrates that our framework can orthogonally benefit from DA techniques, where it is highly uncertain yet for the gradient-based methods. (3) Both FedAvg and DP-FedAvg perform better than their DA variants; therefore we will only use DP-FedAvg in the following experiments. This result is well expected, as FL with domain adaptation is more closely related to the multi-source domain adaptation [Peng et al., 2019a]. Combining *FedAvg* with the one-source DA methods implies averaging different trajectories towards the server's distribution, which may not work in practice. Similar learning bound based observation has been investigated in Peng et al. [2019b] and it remains unclear how to privatize the multi-source domain adaptation approach. On the other hand, leveraging the majority vote is more stable against the distribution shift. We conjecture this is because whenever there is a high consensus among the vote counts, the returned label remains unchanged if the distribution of some agents is slightly perturbed. In contrast, averaging trajectories in such case may diverge the optimization procedure directly.

## 4.2 Instance-level DP Evaluation

We investigate the instance-level DP using datasets Office-Caltech10 [Gong et al., 2012] to further highlight that our method can facilitate to the extreme challenging domain shift scenario, while not explicitly applying any of the domain adaptation technique. Office-Caltech consists of data from four domains: Caltech (C), Amazon (A), Webcam(W) and DSLR (D). We iteratively pick one domain as the server domain each time and the rest ones are for local agents (e.g., in $A, C, D \rightarrow W$, W is treated as the server). For *kNN-DPFL*, we instantiate the public feature extractor using the network backbone without the classifier layer. The DP-FedSGD method provides the DP baseline where we use mostly the same parameters as Abadi et al. [2016]. In each experiment, we split 70% data from the server domain as the public available unlabeled data, which is also the data to be labeled for *kNN-DPFL*, while the remaining 30% data is used for testing.

In Table 2, we observe: (1) DP-FedSGD degrades when the backbone changes from the light load AlexNet to the heavy load ResNet50, while ours is improved by 10%. It is because larger model capacity leads to more sensitive response to gradient clipping or noise injection, which has been surveyed in Abadi et al. [2016]. In contrast, our *kNN-DPFL* avoids the gradient operation by label aggregation and can still benefit from the larger model capacity. Again, our method achieves consistently better utility-privacy trade-off as maintaining same privacy cost and can achieve significantly better utility, or maintaining same utility and can achieve much lower privacy cost.

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

## A  Challenges in Gradient-based Federated Learning

In this section, before introducing our approaches, we motivate them by highlighting the main challenges in the conventional DPFL methods in terms of gradient estimation, convergence, and data heterogeneity. For other challenges, we refer the readers to a survey [Kairouz et al., 2019].

**Challenge 1: Biased Gradient Estimation.** Recent works [Li et al., 2018] have shown that the FedAvg may not converge well under data heterogeneity. We provide a simple example to show that the clipping step of DP-FedAvg may exacerbate the issue.

**Example 5.** *Let $N = 2$, each agent $i$'s local update is $\triangle_i$ ($E$ iterations of SGD). We enforce clipping of per-agent update $\triangle_i$ by performing $\triangle_i / \max(1, \frac{||\triangle_i||_2}{S})$, where $S$ is the clipping threshold. Consider the special case when $||\triangle_1||_2 = S + \alpha$ and $||\triangle_2||_2 \leq S$. Then the global update will be $\frac{1}{2}(\frac{S\triangle_1}{||\triangle_1||_2} + \triangle_2)$, which is biased.*

Comparing to the FedAvg updates $\frac{1}{2}(\triangle_1 + \triangle_2)$, the biased update could be 0 (not moving) or pointing towards the opposite direction. Such a simple example can be embedded in more realistic problems, causing substantial bias that leads to non-convergence.

**Challenge 2: Slow Convergence.** Following works on FL convergence analysis [Li et al., 2019, Wang et al., 2019], we derive the convergence analysis on DP-FedAvg and demonstrate that using many outer-loop iterations ($T$) could result in similar convergence issue under differential privacy.

The appeal of FedAvg is to set $E$ to be larger so that each agent performs $E$ iterations to update its own parameters before synchronizing the parameters to the global model, hence reducing the number of rounds in communication. We show that the effect of increasing $E$ is essentially increasing the learning rate for a large family of optimization problems with piece-wise linear objective functions, which does not change the convergence rate. The detailed analysis is in appendix convergence section due to space limit. Specifically, it is known that for the family of $G-$Lipschitz functions supported on a $B$-bounded domain, any Krylov-space method [1] has convergence rate that is lower bounded by $\Omega(BG/\sqrt{T})$ [Nesterov, 2003, Section 3.2.1]. This indicates that the variant of FedAvg requires $\Omega(1/\alpha^2)$ rounds of outer loop (i.e., communication), in order to converge to an $\alpha$ stationary point, i.e., increasing $E$ does *not* help, even if no noise is added.

It also indicates that DP-FedAvg is essentially the same as *stochastic* sub-gradient method in almost all locations of a piece-wise linear objective function with gradient noise being $\mathcal{N}(0, \sigma^2/NI_d)$. The additional noise in DP-FedAvg imposes more challenges to the convergence. If we plan to run $T$ rounds and achieve $(\epsilon, \delta)$-DP, we need to choose $\sigma = \frac{\eta EG\sqrt{2T\log(1.25/\delta)}}{N\epsilon}$ [McMahan et al., 2018, Theorem 1], which results in a convergence rate upper bound of $\frac{GB(\sqrt{1 + \frac{2Td\log(1.25/\delta)}{N^2\epsilon^2}})}{\sqrt{T}} = O\left(\frac{GB}{\sqrt{T}} + \frac{\sqrt{d\log(1.25/\delta)}}{N\epsilon}\right)$, for an optimal choice of the learning rate $E\eta$.

The above bound is tight for stochastic sub-gradient methods, and in fact also information-theoretically optimal. The $GB/\sqrt{T}$ part of upper bound matches the information-theoretical lower bound for all methods that have access to $T$-calls of stochastic sub-gradient oracle [Agarwal et al., 2009, Theorem 1]. While the second matches the information-theoretical lower bound for all $(\epsilon, \delta)$-differentially private methods on the agent level [Bassily et al., 2014, Theorem 5.3]. That is, the first term indicates that there must be *many rounds of communications*, while the second term says that the *dependence in ambient dimension $d$* is unavoidable for DP-FedAvg. Clearly, our method also has such dependence *in the worst case*. But it is easier for our approach to adapt to the structure that exists in the data (i.e., high consensus among voting), as we will illustrate later. In contrast, it has larger impact on

---

[1] One that outputs a solution in the subspace spanned by a sequence of sub-gradients.

DP-FedAvg, since it needs to explicitly add noise with variance $\Omega(d)$. Another observation is when $N$ is small, no DP method with reasonable $\epsilon, \delta$ parameters is able to achieve high accuracy for agent-level DP. This partially motivates us to consider the other regime that deals with instance-level DP.

**Challenge 3: Data Heterogeneity.** Federated learning with domain adaptation has been studied in Peng et al. [2019b], where they propose a dynamic attention model to adjust the contribution from each source (agent) collaboratively. However, most multi-source domain adaptation algorithms, including this approach, require sharing local feature vectors to the target domain, which is not compatible with the DP setting. Enhancing *DP-FedAvg* with the effective domain adaptation technique remains an open problem.

# B    Other properties of differential privacy

**Definition 6** (Renyi Differential Privacy [Mironov, 2017]). *We say a randomized algorithm $\mathcal{M}$ is $(\alpha, \epsilon(\alpha))$-RDP with order $\alpha \geq 1$ if for neighboring datasets $D, D'$,*

$$\mathbb{D}_\alpha(\mathcal{M}(D)||\mathcal{M}(D')) := \frac{1}{\alpha - 1} \log \mathbb{E}_{o \sim \mathcal{M}(D')} \left[ \left( \frac{\Pr[\mathcal{M}(D) = o]}{\Pr[\mathcal{M}(D') = o]} \right)^\alpha \right] \leq \epsilon(\alpha).$$

RDP inherits and generalizes the information-theoretical properties of DP.

**Lemma 7** (Selected Properties of RDP [Mironov, 2017]). *If $\mathcal{M}$ obey $\epsilon_\mathcal{M}(\cdot)$-RDP, then*

1. *[Indistinguishability] For any measurable set $S \subset Range(\mathcal{M})$, and any neighboring $D, D'$*

$$e^{-\epsilon(\alpha)}\Pr[\mathcal{M}(D') \in S]^{\frac{\alpha}{\alpha-1}} \leq \Pr[\mathcal{M}(D) \in S] \leq e^{\epsilon(\alpha)}\Pr[\mathcal{M}(D') \in S]^{\frac{\alpha-1}{\alpha}}.$$

2. *[Post-processing] For all function $f$, $\epsilon_{f \circ \mathcal{M}}(\cdot) \leq \epsilon_\mathcal{M}(\cdot)$.*

3. *[Composition] $\epsilon_{(\mathcal{M}_1, \mathcal{M}_2)}(\cdot) = \epsilon_{\mathcal{M}_1}(\cdot) + \epsilon_{\mathcal{M}_2}(\cdot)$.*

This composition rule often allows for tighter calculations of $(\epsilon, \delta)$-DP for the composed mechanism than the strong composition theorem in [Kairouz et al., 2015]. Moreover, we can covert RDP to $(\epsilon, \delta)$-DP for any $\delta > 0$ using:

**Lemma 8** (From RDP to DP). *If a randomized algorithm $\mathcal{M}$ satisfies $(\alpha, \epsilon(\alpha))$-RDP, then $\mathcal{M}$ also satisfies $(\epsilon(\alpha) + \frac{\log(1/\delta)}{\alpha-1}, \delta)$-DP for any $\delta \in (0, 1)$.*

**Threat models and Multi-Party Computation (MPC)**  However, the privacy guarantee of DP-FedAvg only applies to the global model and does not apply to the inference made by curious parties who can eavesdrop in the network traffics. Cryptographic techniques such as Multi-Party Computation (MPC) [Yao, 1982] securely aggregates local updates and ensures privacy against inferences made during the communication process. Specifically, if each party adds a small independent noise to the part they contribute, MPC ensures that an attacker can only observe the total, even if he taps the network messages and hacks into the server. Unfortunately, it is challenging to apply MPC in either DP-FedAvg or DP-FedSGD due to high computational overheads. As shown in Bonawitz et al. [2017a], the computational cost of security aggregation (used as MPC-Sum in Figure 1) is $O(N^2 + dN)$ for users and $O(dN^2)$ for the server, where $d$ is the model size and $N$ is the number of agents. In this paper, we consider a new MPC technique due to [Dery et al., 2019] that allows only the voted winner to be released while keeping the voting scores completely hidden. This allows us to further amplify the DP guarantees. In our experiment, we assume the aggregation is conducted by MPC for all privacy-preserving algorithms that we consider (see Figure 1).

# C    More Discussions of Challenges for Gradient-Based FL

**Definition 9.** *A function $\ell$ is Lipschitz continuous with constant $G > 0$, if*

$$|\ell(x) - \ell(y)| \leq G||x - y||_2$$

*for all $x, y$.*

**Proposition 10.** *Let the objective function of agents $f_1, ..., f_N$ obeys that $f_i$ is piecewise linear (which implies that the global objective $F = \frac{1}{N}\sum_{i=1}^{N} f_i$ is piecewise linear) and $G$-Lipschitz. Let $\eta$ be the learning rate taken by individual agents. Then the outer loop FedAvg update is equivalent to $\theta^+ = \theta - E\eta g$ for some $g \in \mathbb{R}^d$, where (a) $g = \nabla F(\theta)$ if $\theta$ is in the $\nu$ interior of the linear region of $f_1, ..., f_N$ and $E < \nu/(\eta G)$; (2) $g$ is a Clarke-subgradient [2] of $F$ at $\theta$, if $\theta$ is on the boundary of at least two linear regions and at least $\nu$ away in Euclidean distance from another boundary and $E < \nu/(\eta G)$; (c) otherwise, we have that $\|g - \nabla F(\theta)\|_2 \leq E\eta G$. Moreover, statement (c) is true even if we drop the piecewise linear assumption.*

*Proof.* For the Statement (a), observe that for all $\theta'$ such that $\|\theta' - \theta\| \leq \nu$ neighborhood, we have that $\nabla f_i(\theta') = \nabla f_i(\theta)$. When $E < \nu/(\eta G)$, the cumulative gradients of agent $i$ is equal to $E\nabla f_i(\theta)$. For Statement (b), notice that the Clarke subdifferential at $\theta$ is the convex hull of the one-sided gradient, thus as we move along the negative gradient direction in the inner loop, we enter and remains in the linear region. Thus the update direction is

$$\frac{1}{N}\left( \sum_{i \text{ s.t. } f_i \text{ is differentiable at } \theta} E\eta\nabla f_i(\theta) + \sum_{i \text{ s.t. } f_i \text{ is not differentiable at } \theta} \eta g_i + (E-1)\nabla f_i(\theta - \eta g_i) \right)$$

for all $g_i$ such that it is a Clarke-subgradient of $f_i$ it can be written as a convex combination. The proof is complete by observing that the $1/N \sum_i$ is also a convex combination and by multiplying and dividing by $E$. Statement (c) is a straightforward application of the Lipschitz property which says that $E$ steps can at most get you away for $\eta E G$ and clearly piecewise linear assumption is not required. $\qquad\square$

This proposition says that in almost all $\theta$, increasing $E$ has the effect of increasing the learning rate of the subgradient "descent" method for piecewise linear objective functions; and increasing the learning rate of an approximate gradient method in general for Lipschitz objective functions. It is known that for the family of $G-$Lipschitz function supported on a $B$-bounded domain, any Krylov-space method [3] has a rate of convergence that is lower bounded by $O(BG/\sqrt{T})$ if running for $T$ iterations. A close inspection of the lower bound construction reveals that the worst-case problem is $\min_{\theta \in \mathbb{R}^T} \max_i \theta_i + \|\theta\|^2$, namely, a regularized piecewise linear function. This is saying that the variant of FedAvg that aggregates only the loss-function part of the gradient or projects only when synchronizing essentially requires $\Omega(1/\alpha^2)$ rounds of outer loop iterations (thus communication) in order to converge to an $\alpha$ stationary point, i.e., increasing $E$ does *not* help, even if no noise is added.

**Lemma 11** (Restatement of Lemma **??**). *Conditioning on the teachers, for each public data point $x$, the noise added to each coordinate is drawn from $\mathcal{N}(0, \sigma^2/N^2)$, then with probability $\geq 1 - C\exp\{-N^2\gamma(x)^2/8\sigma^2\}$, the privately released label matches the majority vote without adding noise.*

*Proof.* The proof is a straightforward application of Gaussian tail bounds and a union bound over $C$ coordinates. Specifically, $\mathbb{P}[Z_{j^*} < -\gamma(x)/2] \leq e^{-\frac{N^2\gamma(x)^2}{8\sigma^2}}$ for the argmax $j^*$. For $j \neq j^*$, $\mathbb{P}[Z_j > \gamma(x)/2] \leq e^{-\frac{N^2\gamma(x)^2}{8\sigma^2}}$. By a union bound over all coordinates $C$, we get that there perturbation from the boundedness is smaller than $\gamma(x)/2$, which implies correct release of the majority votes. $\qquad\square$

This lemma implies that for all public data point $x$ such that $\gamma(x) \geq \frac{2\sqrt{2\log(C/\delta)}}{N}$, the output label matches noiseless majority votes with probability exponentially close to 1.

# D  Data-dependent Privacy Analysis

---

[2]Clarke-subgradient is a generalization of the subgradient to non-convex functions. It reduces to the standard (Moreau) subgradient when $F$ is convex.

[3]One that outputs a solution in the subspace spanned by a sequence of subgradients.

## D.1 Privacy Analysis

**Theorem 12** (Restatement of Theorem 4). *Let AE-DPFL and kNN-DPFL answer $Q$ queries with noise scale $\sigma$. For agent-level protection, both algorithms guarantee $(\alpha, Q\alpha/(2\sigma^2))$-RDP for all $\alpha \geq 1$. For instance-level protection, AE-DPFL and kNN-DPFL obey $(\alpha, Q\alpha/\sigma^2)$ and $(\alpha, Q\alpha/(k\sigma^2))$-RDP respectively.*

*Proof.* In *AE-DPFL*, for query $x$, by the independence of the noise added, the noisy sum is identically distributed to $\sum_{i=1}^{N} f_i(x) + \mathcal{N}(0, \sigma^2)$. Adding or removing one data instance from will change $\sum_{i=1}^{N} f_i(x)$ by at most $\sqrt{2}$ in L2. The Gaussian mechanism thus satisfies $(\alpha, \alpha s^2/2\sigma^2)$-RDP on the instance-level for all $\alpha \geq 1$ with an L2-sensitivity $s = \sqrt{2}$. This is identical to the analysis in the original PATE [Papernot et al., 2018].

For the agent-level, the L2 and L1 sensitivities are both $1$ for adding or removing one agent.

In *kNN-DPFL*, the noisy sum is identically distributed to $\frac{1}{k}\sum_{i=1}^{N}\sum_{j=1}^{k} y_{i,j} + \mathcal{N}(0, \sigma^2)$. The change of adding or removing one agent will change the sum by at most $1$, which implies the same L2 sensitivity and same agent-level protection as *AE-DPFL*. The $L2$-sensitivity from adding or removing one instance, on the other hand changes the score by at most $\sqrt{2/k}$ in L2 due to that the instance being replaced by another instance, this leads to an an improved instance-level DP that reduces $\epsilon$ by a factor of $\sqrt{\frac{k}{2}}$.

The overall RDP guarantee follows by the composition over $Q$ queries. The approximate-DP guarantee follows from the standard RDP to DP conversion formula $\epsilon(\alpha) + \frac{\log(1/\delta)}{\alpha-1}$ and optimally choosing $\alpha$. $\qquad\square$

## D.2 Improved accuracy and privacy with large margin

Let $f_1, ..., f_N : \mathcal{X} \to \triangle^{C-1}$ where $\triangle^{C-1}$ denotes the probability simplex — the soft-label space. Note that both algorithms we propose can be viewed as voting of these teachers which outputs a probability distribution in $\triangle^{C-1}$. First let us define the margin parameter $\gamma(x)$ which measures the difference between the largest and second largest coordinate of $\frac{1}{N}\sum_{i=1}^{N} f_i(x)$.

**Lemma 13.** *Conditioning on the teachers, for each public data point $x$, the noise added to each coordinate is drawn from $\mathcal{N}(0, \sigma^2/N^2)$, then with probability $\geq 1 - C\exp\{-N^2\gamma(x)^2/8\sigma^2\}$, the privately released label matches the majority vote without adding noise.*

*Proof.* The proof is a straightforward application of Gaussian tail bounds and a union bound over $C$ coordinates. Specifically, $\mathbb{P}[Z_{j^*} < -\gamma(x)/2] \leq e^{-\frac{N^2\gamma(x)^2}{8\sigma^2}}$ for the argmax $j^*$. For $j \neq j^*$, $\mathbb{P}[Z_j > \gamma(x)/2] \leq e^{-\frac{N^2\gamma(x)^2}{8\sigma^2}}$. By a union bound over all coordinates $C$, we get that there perturbation from the boundedness is smaller than $\gamma(x)/2$, which implies correct release of the majority votes. $\qquad\square$

This lemma implies that for all public data point $x$ such that $\gamma(x) \geq \frac{2\sqrt{2\log(C/\delta)}}{N}$, the output label matches noiseless majority votes with probability exponentially close to $1$.

Next we show that for those data point $x$ such that $\gamma(x)$ is large, the privacy loss for releasing $\text{argmax}_j[\frac{1}{N}\sum_{i=1}^{N} f_i(x)]_j$ is exponentially smaller. The result is based on the following privacy amplification lemma that is a simplification of Theorem 6 in the appendix of [Papernot et al., 2018].

**Lemma 14.** *Let $\mathcal{M}$ satisfy $(2\alpha, \epsilon)$-RDP, and there is a singleton output that happens with probability $1-q$ when $\mathcal{M}$ is applied to $D$. Then for any $D'$ that is adjacent to $D$, Renyi-divergence*

$$D_\alpha(\mathcal{M}(D)\|\mathcal{M}(D')) \leq -\log(1-q) + \frac{1}{\alpha-1}\log(1 + q^{1/2}(1-q)^{\alpha-1}e^{(\alpha-1)\epsilon}).$$

*Proof.* Let $P, Q$ be the distribution of of $\mathcal{M}(D)$ and $\mathcal{M}(D')$ respectively and $E$ be the event that the singleton output is selected.

$$
\begin{aligned}
\mathbb{E}_Q[(dP/dQ)^\alpha] &= \mathbb{E}_Q[(dP/dQ)^\alpha | E]\mathbb{P}_Q[E] + \mathbb{E}_Q[(dP/dQ)^\alpha \mathbf{1}(E^c) \\
&\leq (1-q)(\frac{1}{1-q})^\alpha + \sqrt{\mathbb{E}_Q[(dP/dQ)^{(2\alpha)}]}\sqrt{\mathbb{E}_Q[\mathbf{1}(E^c)^2]} \\
&\leq (1-q)^{-(\alpha-1)} + q^{1/2}e^{(2\alpha-1)\epsilon/2} = (1-q)^{-(\alpha-1)}\left(1 + (1-q)^{\alpha-1}q^{1/2}e^{\frac{2\alpha-1}{2}\epsilon}\right)
\end{aligned}
$$

The first part of the second line uses the fact that event $E$ is a singleton with probability larger than $1-q$ under $Q$ and the probability is always smaller than $1$ under $P$. The second part of the second line follows from Cauchy-Schwartz inequality. The third line substitute the definition of $(2\alpha, \epsilon)$-RDP. Finally, the stated result follows by the definition of the Renyi divergence. $\qquad\square$

**Theorem 15** (Restatement of Theorem **??**). *The mechanism that releases* $\mathrm{argmax}_j[\frac{1}{N}\sum_{i=1}^N f_i(x) + \mathcal{N}(0, (\sigma^2/N^2)I_C)]_j$ *obeys* $(\alpha, \epsilon)$-*data-dependent-RDP, where*

$$
\epsilon \leq 2Ce^{-\frac{N^2\gamma(x)^2}{8\sigma^2}} + \frac{1}{\alpha-1}\log\left(1 + e^{\frac{(2\alpha-1)\alpha s}{2\sigma^2} - \frac{N^2\gamma(x)^2}{8\sigma^2} + \log C/2}\right),
$$

*where $s = 1$ for AE-DPFL with the agent-level DP, and $s = 2/k$ for KNN-DPFL with the instance-level DP.*

*Proof.* The proof involves substituting $q = Ce^{-\frac{N^2\gamma(x)^2}{8\sigma^2}}$ from Lemma **??** into Lemma 14 and use the fact that $\mathcal{M}$ satisfies the RDP of a Gaussian mechanism from the RDP's post-processing lemma. The expression bound is simplified for readability using $-\log(1-x) < 2x$ for all $x > -0.5$ and that $(1-q)^{\alpha-1} \leq 1$. $\qquad\square$

As we can see, when given teachers that are largely in consensus, the (data-dependent) privacy loss exponentially smaller.

