# OpenReview forum: "VOTING-BASED APPROACHES FOR DIFFERENTIALLY PRIVATE FEDERATED LEARNING"
_NeurIPS.cc/2022/Workshop/Federated_Learning — FL-NeurIPS 2022 Poster_

### Official Review · Reviewer_9qk4 · 2022-10-04
**The manuscript mainly studies differentially private federated learning. The authors design two algorithms to guarantee both instance-level and agent-level privacy. In order to reduce the communication cost, a voting scheme is proposed instead of averaging the gradients. The secure multi-party computation could exponentially amplify the privacy guarantees.**

The manuscript mainly studies differentially private federated learning. The authors design two algorithms to guarantee both instance-level and agent-level privacy. In order to reduce the communication cost, a voting scheme is proposed instead of averaging the gradients. The secure multi-party computation could exponentially amplify the privacy guarantees. Detailed comments are listed as follows.

1. What's the difference between differentially private and differential privacy?
2. Many existing works have focused on differential privacy-based FL, but this paper doesn’t mention them in the related works.
3. From the existing description in the article, it is impossible to see the execution order of algorithms DP-FedSGD and kNN-DPFL.
4. In the kNN-DPFL, how to set the number of k? Who is responsible for the algorithm execution?
5. MPC protocols approach can reduce communication overheads, but the rationale behind it is not clear.
6. In the MPC-vote, how to calculate the voting score?
7. In the experimental part, it is recommended to add the display of visualized results.

---

### Official Review · Reviewer_wXKC · 2022-10-12
**Voting-based Approaches for Differentially Private Federated Learning**

This paper presents two student-teacher ensemble based (voting)
approaches, AE-DPFL and kNN-DPFL, for federated learning derived from
prior work on PATE and and Private-KNN.  The original works were
applied to settings that were close to, but not quite the same as the
FL setting.  This paper appears to have retrofitted those algorithms
in the FL setting.  For additional privacy, the authors add a secret
sharing based MPC layer for secure aggregation of the votes.  Their
algorithms also provide agent- and instance-level Differential Privacy
guarantees.  Empirical results show both their algorithms
outperforming a DPSGD-based FL algorithm.

The main issue with the paper is that the work appears to be a
derivative of existing work, and very incremental.  However, their
combination with MPC based secure aggregation for stronger privacy,
and the algorithms' support for instance- and agent-level privacy
appear to be somewhat novel.  Empirical evaluation also appears to
show promise for these algorithms.

Additional comments/suggestions:

* Formal treatment on utility loss compared to FedAvg would make the
paper stronger.

* A deeper dive into the effects of decisions made in local training at
each agent on the ensemble model at the server could be quite
enlightening.

* If possible, the authors should try to quantify the communication
overheads in their empirical evaluation section.

* I having a conclusion at the end of the paper gives the authors a good
means to sum up the key takeaways for the reader.

---

### Official Review · Reviewer_LaYA · 2022-10-17

This paper proposed to use voting technique to avoid dimension dependence and reduce the communication cost. This paper proposed two FL algorithms with differential privacy guarantees for both instance-level and agent-level privacy regimes. This paper conducted extensive experiments to verify the effectiveness of their proposed methods. This paper showed that their methods can improve the tradeoff between privacy and model utility.

My comments are as following:

1. a formal definition of instance-level and agent-level dp is needed.

2. a description of the threat model in this paper is needed.

3. a conclusion section is needed to talk about the limitations and future work.

---

### Decision · Program_Chairs · 2022-10-20

Accept (Poster)